# A Cross-Sectional Study of Knowledge, Attitudes, and Practices concerning COVID-19 Outbreaks in the General Population in Malang District, Indonesia

**DOI:** 10.3390/ijerph19074287

**Published:** 2022-04-03

**Authors:** Sujarwoto Sujarwoto, Holipah Holipah, Asri Maharani

**Affiliations:** 1Portsmouth Brawijaya Center for Global Health, Population and Policy & Department of Public Administration, Universitas Brawijaya, Malang 65145, Indonesia; 2Portsmouth Brawijaya Center for Global Health, Population and Policy & Faculty of Medicine, Universitas Brawijaya, Malang 65142, Indonesia; holifah.fkub@ub.ac.id; 3Division of Nursing, Midwifery & Social Work, University of Manchester, Manchester M13 9PL, UK; asri.maharani@manchester.ac.uk

**Keywords:** COVID-19, KAP, cross-sectional study, scarce health resources, district, Indonesia

## Abstract

Lack of knowledge often leads to nonchalant attitudes and improper practices that expose people to greater risks during a pandemic. Therefore, improving the general public’s knowledge, attitudes, and practices (KAP) concerning coronavirus disease (COVID-19) can play a pivotal role in reducing the risks, especially in a country such as Indonesia with its scarcity of health resources for testing and tracing. Using the case of Malang District, this study set out to evaluate KAP regarding COVID-19 and its risk factors immediately after the Malang health authorities implemented various preventive measures. A population-based survey involving 3425 individuals was carried out between 1 May and 20 May 2020. Our findings revealed that less than half of the respondents demonstrated accurate knowledge (25.3%), positive attitudes (36.6%), or frequent best practices (48.8%) with regard to COVID-19 prevention. The results of logistic regression analyses showed that more accurate knowledge was associated with more positive attitudes and more frequent best practices (OR = 1.603, *p*-value < 0.001; OR = 1.585, *p*-value < 0.001, respectively). More positive attitudes were also associated with more frequent best practices (OR = 1.126, *p*-value < 0.001). The level of KAP varied according to sociodemographic characteristics, access to the services of community health workers, and mobile health technology for COVID-19 screening. Some global health proposals to improve health behaviors among the general public in the context of the scarcity of health resource settings are suggested based on the study findings.

## 1. Introduction

Improving the general public’s knowledge regarding COVID-19 is important in order to reduce the pandemic’s risks and suppress coronavirus transmission within society. However, experience from past epidemics shows that a lack of knowledge often leads to nonchalant attitudes and improper practices that expose people to greater risks [1]. For example, evidence during the SARS pandemic in China in 2003 shows that a lack of knowledge to the disease was linked to panic attacks and emotional reactions among citizens, which complicated the authorities’ efforts to control the virus’s spread [1].

Several COVID-19 KAP studies suggest that KAP intervention is one of the key public health strategies for controlling diseases by changing citizens’ health behaviors [2]. However, existing empirical studies have reported mixed findings [3]. A study in Bangladesh found that people with more knowledge of COVID-19 were more likely to have more positive attitudes and to engage in prevention practices [3]. Other research revealed that knowledge about COVID-19 was insufficient to prompt behavioral change among Ecuadorians [4]. A study in Saudi Arabia reported a significant relationship between knowledge and practices, but the association was weak [5]. The influence of KAP intervention related to COVID-19 prevention practices depended not only on the type of intervention but also on various sociodemographic factors [6]. According to a study conducted in Hubei, China, government actions stemming from the outbreak were strongly linked to perceived risks and awareness of COVID-19 [7]. Furthermore, attitudes toward COVID-19 preventive practices were linked to effective health education strategies [8]. Studies have highlighted the importance of trust and networking for effective public health education [9,10,11]. A study in India also reported the benefits of mobile apps for public health education during the pandemic [11]. These mixed findings suggest a need for further investigation of factors associated with COVID-19 KAP within various contexts in order to better understand the mechanisms by which KAP in the general public can be improved.

Like many other countries, Indonesia’s government has taken unprecedented preventive measures to control the rapid spread of COVID-19, including shutting down government offices, closing schools, supermarkets, hotels, and restaurants, limiting gatherings in mosques and churches, and implementing penalties on gatherings [12]. Various health promotion strategies have also been applied, including developing national health insurance mobile health technology for COVID-19 self-screening and public health education [13]. Moreover, the government has deployed community health workers to support health authorities in community health education [14]. Hence, the purpose of this study was to evaluate KAP regarding COVID-19 prevention after the health authorities’ implementation of various preventive measures to improve public knowledge and to change behaviors, focusing on a district with a significant scarcity in health facilities. Studies suggest that evaluating the extent of citizens’ COVID-19 KAP can help health authorities to identify knowledge gaps and behavioral patterns among sociodemographic subgroups, enabling the design of appropriate health promotion and prevention strategies [15]. In the next section, we explain the methods used in this study, starting with the study’s setting.

## 2. Materials and Methods

### 2.1. Study Setting

This research was carried out in the Malang District of East Java, Indonesia. The second-largest district in East Java Province, Malang, has 3535 square kilometres. The population of Malang District consists of 2,967,315 individuals. It was chosen for this study because of several unique characteristics. The district represents the general condition of the country’s health care system, with limited health resources and facilities to handle the outbreak of COVID-19. Malang has 39 primary health centers, or *Pusat Kesehatan Masyarakat* (*Puskesmas*) (1 per ~65,000 individuals) and 390 village health clinics, or *Pondok Kesehatan Desa* (*Ponkesdes*) (1 per ~7000 individuals) in which community health workers, or *kader*, play pivotal roles in providing community-based health education and preventive care [16]. The district has two COVID-19 hospitals with 553 COVID-19 beds (1 per ~5400 individuals). Health insurance coverage in Malang in 2021 was only 32% [16]. Of the population in Malang District, 42.9% is considered “poor or near poor”, compared to 51% in East Java overall [16]. By 27 March 2022, about 481.4 million people across the globe and 5.9 million Indonesian were infected with COVID-19 [17]. In Indonesia, more than 73.2% of them were from Java, the most densely populated island in the archipelago [18].

### 2.2. Study Design

This study used a cross-sectional design with a population-based sample. Purposive sampling was used to determine the sampling population, where the population was stratified into 39 primary healthcare center areas capturing rural and urban areas [19]. Initially, 3600 respondents in the study sample provided written informed consent. Of these, 3425 respondents completed the entire survey, generating a response rate of 95.1%. The survey was carried out from 1 May to 20 May 2020, two months after the health authorities implemented various preventive measures. A questionnaire was designed for the Kobo Toolbox (a simple, robust, and powerful data collection tool) [20]. The survey apps were used by 39 trained field researchers responsible for collecting data.

### 2.3. Measures

#### 2.3.1. Knowledge, Attitudes, and Practices regarding COVID-19 Prevention

KAP regarding COVID-19 prevention were measured using a World Health Organization (WHO) questionnaire on COVID-19 prevention and control, as well as questionnaires on viral epidemics related to Middle East respiratory syndrome coronavirus (MERS-CoV) [21]. Table 1 describes each question and coding structure for measuring KAP in this study.

The COVID-19 KAP questionnaire was verified and implemented once the original draft had been completed. The verification steps of the questionnaire were as follows: First, we submitted the questionnaire to three academic experts with expertise in the field. After all experts approved it, the final questionnaire was drafted and tested on 30 individuals to ensure reliability. The Cronbach’s alpha coefficients of knowledge, attitudes, and practices in the pilot data were 0.81, 0.82, and 0.84, respectively. The overall Cronbach’s alpha was 0.82, which indicates acceptable internal consistency [22].

#### 2.3.2. Sociodemographic and Other Factors

Based on prior studies, we included several sociodemographic factors associated with KAP regarding COVID-19 [3,23,24,25]. These sociodemographic factors include age, sex, education, type of employment, monthly family income, marital status, rural/urban residence, and family type.

We classified age into three categories: young adult (17–30 years, code = 0), middle adult (31–45, code = 1), and senior adult (>45, code = 2). We used the young adult category as a reference group. Sex was classified into female (code = 1) and male (code = 0, reference group). Formal education was categorized into three groups: elementary school or less (code = 2), junior secondary school (code = 1), and high school or more (code = 0, reference group). Marital status was classified as single (code = 1), divorced (code = 2), widowed (code = 3), or married (code = 0, reference group). The type of employment was classified as civil servant (code = 1), laborer (code = 2), private company worker, i.e., bank, supermarket, hotel, or restaurant (code = 3), farmer (code = 4), trader (code = 5), or student (code = 0 as a reference group). Family type was classified as nuclear family (code = 0 as a reference group) or joint family (code = 1). Monthly family income was categorized into three groups: <1 million Indonesian Rupiah (IDR) (code = 2), 1–3 million (code = 1), and >3 million (code = 0, reference group). Residential area was categorized as urban (code = 0, reference group) or rural (code = 1).

We also included mobile health app use and access to COVID-19-related information from community health workers. A binary variable was created to measure whether respondents used the mobile app (1 = used, 0 = did not use, reference group). In the survey, each respondent was asked: “Did you use the JKN mobile app for COVID-19 self-screening, or have you accessed information related to COVID-19 during the last month?”. A binary variable was created to measure respondent access to COVID-19-related information from community health workers (1 = yes, 0 = no, reference group). Each respondent was asked: “Have you received information from community health workers (*kader*) about COVID-19 prevention or other COVID-19-related information during the last month?”

### 2.4. Statistical Analysis

Descriptive statistics were presented using frequencies and percentages. Multiple logistic regression was performed to quantify the associations between KAP and sociodemographic factors, residential area, mobile health app utilization, and access to COVID-19 information from community health workers. All results were reported in odds ratios (OR) and 95% confidence intervals (95% CI). The data analysis was performed using STATA 17.1 software (StataCorp LLC, College Station, TX, USA).

## 3. Results

### 3.1. Sociodemographic Characteristics and Other Factors

Table 2 shows sociodemographic characteristics and KAP regarding COVID-19 drawn from the questionnaire. Most respondents were female (62.4%), and the largest age group was that of middle-aged adults (45.7%). About half of the respondents had an elementary school education or less (51.5%), and 82.3% had an income of less than 1 million IDR per month (~70 United States Dollar or USD). Most respondents were married (73.2%), and 51.5% worked as farmers. Most lived within a nuclear family (63.8%) and in a rural area (97.1%). Only 7.2% reported using the JKN COVID-19 mobile app for COVID self-screening, and 24.2% reported having received information regarding COVID-19 from community health workers in the past month. Of the respondents, 25.3% correctly answered four questions regarding COVID-19, 36.6% were classified as having positive attitudes, and 48.8% reported following frequent best practices.

### 3.2. Knowledge, Attitudes and Practices regarding COVID-19

Table 3 presents each question concerning the knowledge, attitudes, and practices of 3425 respondents. Most respondents appeared to know that COVID-19 is a dangerous disease (94.9% answered correctly). However, most had misunderstandings about COVID-19, such as believing that infection could occur through animal products (30.3%), having contact with wild animals (46.9%), and through consuming well-cooked products (21.3%). With regard to attitudes, most respondents agreed that it was necessary to report a suspected case to health authorities (93.7%), to use a face mask in a crowded place (97.2%), and to wash the hands and face after being outside (96.5%). Furthermore, most believed COVID-19 to be a preventable disease (92.0%) and agreed that health education could play an important role in prevention (93.4%). However, only 69.7% of respondents agreed that COVID-19 patients could be treated at home. In terms of practices regarding COVID-19, 72.1% reported that they washed their hands frequently using water and soap. Most reported maintaining a healthy lifestyle during the pandemic (69.3%). However, misunderstandings surfaced through questionnaire items that asked whether respondents avoided touching face and eyes (51.8%), maintained social distance or quarantined at home (57.3%), used tissues or handkerchiefs when they coughed or sneezed (57.9%), and obeyed all government rules related to COVID-19 prevention (57.9%).

### 3.3. Multiple Logistic Regression Results

Figure 1 shows multiple logistic regression analyses for knowledge about COVID-19. All odds ratios are reported. The odds of having more accurate knowledge were 1.641 times higher among females than among males (*p*-value ≤ 0.001). Middle-aged and senior adults were one and one and a half times more likely to have more accurate knowledge than young adults (odds ratio or OR = 1.577, *p*-value ≤ 0.001 for middle-aged adults and OR = 1.557, *p*-value = 0.003 for senior adults). As expected, individuals who were educated to elementary or junior secondary school levels had less accurate knowledge about COVID-19 than those educated to a high school level or higher (OR = 0.492, *p*-value = 0.001 for elementary school and OR = 0.540, *p*-value ≤ 0.001 for junior secondary school).

No significant associations appeared between economic status and knowledge. The odds of having less accurate knowledge for divorced and widowed individuals were greater than for married individuals (OR = 0.273, *p*-value = 0.004 for divorced individuals, OR = 0.452 and *p*-value = 0.001 for widowed individuals). All types of workers (laborers, privately employed workers, farmers, and traders) demonstrated less accurate knowledge than students. However, the relationship appeared statistically significant only for farmers and privately employed workers (OR = 0.800, *p*-value < 0.056 for farmers and OR = 0.661, *p*-value < 0.029 for privately employed workers). The odds of having more accurate knowledge for individuals living within a joint family were almost four times higher than for those living within nuclear families (OR = 3.766, *p*-value ≤ 0.001). There was no significant association between place of residence and knowledge. The odds of having more accurate knowledge among individuals who reported being informed about COVID-19 from community health workers were 1.576 times larger than among those who did not (*p*-value ≤ 0.001). The odds of having more accurate knowledge for individuals who used JKN COVID-19 mobile app screening were 3.632 times greater than for those who did not use the app (OR = 3.632, *p*-value ≤ 0.001).

Figure 2 shows multiple logistic regression results for attitudes toward COVID-19. Adjusting for sociodemographic factors, the odds of having more positive attitudes were 1.603 times higher for individuals with more accurate knowledge than for those with less accurate knowledge (*p*-value ≤ 0.001). Females had 1.438 times higher odds of having more positive attitudes than males (OR = 1.438, *p*-value ≤ 0.001). The odds of middle-aged having positive attitudes were 1.365 times higher than those of young adults (*p*-value = 0.005). However, a non-significant association was found for senior adults. Individuals with an elementary education and junior secondary school had less positive attitudes than individuals with a high school or higher-level education (OR = 0.651, *p*-value ≤ 0.001, OR = 0.446, *p*-value ≤ 0.001). The odds of having more positive attitudes were lower among individuals from higher family income group (>3 million IDR per month) than among individuals from the lower family income group (<1 million IDR per month) (OR = 0.471, *p*-value = 0.046). No significant association was shown for individuals from the 1–3 million IDR income group. Widowed and single individuals had less positive attitudes than married individuals (OR = 0.376, *p*-value = 0.022). The relationship of divorced status on attitudes was not statistically significant. Traders had less positive attitudes than students (OR = 0.739, *p*-value = 0.067). Individuals who lived within joint families had more positive attitudes than individuals who lived within nuclear families (OR = 1.656, *p*-value ≤ 0.001). Individuals who live in rural area having less positive attitude than those live in urban area (OR = 0.512, *p*-value = 0.006). The odds of having more positive attitudes were 1.547 times greater for individuals who reported being informed about COVID-19 by community health workers than for those who did not (*p*-value ≤ 0.001). The association of JKN COVID-19 mobile app screening and attitudes also appears significant (OR = 1.603, *p*-value ≤ 0.001).

Figure 3 shows the results of logistic regression for practices toward COVID-19. Considering sociodemographic factors, the odds of having more frequent best practices were 1.585 higher among individuals who had more accurate knowledge than among individuals who had less accurate knowledge (*p*-value ≤ 0.001). The odds of having more frequent best practices were 1.126 times higher among individuals with more positive attitudes than among those with less positive attitudes (*p*-value ≤ 0.001). Females were more likely to practice COVID-19 prevention than males (OR = 1.419, *p*-value = 0.001). The odds of more frequent best practices for middle-aged adults were 1.351 times greater than for young adults (*p*-value = 0.007). As expected, less formal education was associated with weaker practices regarding COVID-19 prevention (OR = 0.659, *p*-value ≤ 0.001 for elementary education and less; OR = 0.444, *p*-value ≤ 0.001 for junior secondary education). However, individuals from higher family income groups (>3 million IDR per month) engaged less frequently in the best practices than those from lower family income groups (OR = 0.569, *p*-value = 0.013). The odds of engaging in less healthy practices were greater for widowed individuals than for married individuals (OR = 0.550, *p*-value = 0.006). No significant association was shown between single and divorced status and the best practices regarding COVID-19 prevention.

Traders engaged in the best practices less frequently than students (OR = 0.728, *p*-value = 0.054). Other types of employment appeared to be insignificant. The odds of more frequent best practices for individuals living within joint families were 1.677 times higher than for those living within nuclear families (*p*-value ≤ 0.001). Individuals who lived in urban areas were less likely to practice COVID-19 prevention than individuals living in rural areas (OR = 0.501, *p*-value = 0.004). The odds of more frequent best practices were 1.543 times greater for individuals who reported being informed about COVID-19 from community health workers than individuals who did not (*p*-value ≤ 0.001). Likewise, the odds of more frequent best practices were 1.905 times greater for individuals who used JKN COVID-19 mobile app screening than for those who did not use the app (*p*-value ≤ 0.001).

## 4. Discussion

This study was conducted to examine knowledge, attitudes, and practices regarding COVID-19 in Malang District, East Java, Indonesia. In contrast to similar KAP studies in Bangladesh, China, India, Nigeria, and Malaysia, our findings show that most individuals had less than adequate knowledge, less positive attitudes, and less frequent best practices regarding COVID-19 [3,24,25,26,27]. After the health authorities implemented various preventive measures to suppress virus transmission, our findings revealed that fewer than half of respondents demonstrated accurate knowledge (25.3%), positive attitudes (36.6%), or frequent best practices (48.8%). Compared to other recent KAP surveys from Bangladesh, China, India, Nigeria, and Malaysia [3,24,25,26,27], our study uncovered markedly reduced accurate knowledge, positive attitudes, and frequent best practices regarding the disease. For example, a KAP study in Malaysia reported that 83.1% of respondents held positive attitudes toward the successful control of COVID-19 [24]. A study in Nigeria reported 99.5% and 79.5% of respondents had good knowledge and positive attitudes toward COVID-19 prevention [26]. A study in Bangladesh revealed that 48.3% of respondents had more accurate knowledge, 62.3% had more positive attitudes, and 55.1% had more frequent practices regarding COVID-19 prevention [3].

Our study settings and population may have influenced the KAP differences. Our study was based on face-to-face surveys with a majority of respondents from junior secondary school or lower education levels (71.9%), focusing on rural and resource-limited settings, where public information and dissemination regarding COVID-19 is often lacking. In contrast, KAP surveys from Bangladesh, China, India, Nigeria, and Malaysia were based on online surveys with a bias toward respondents with higher education levels and metropolitan populations, where COVID-19 related information is often frequently available. For example, of the participants of the Bangladesh study, 87.8% were from undergraduate or graduate education backgrounds [3]; in the China study, 63.5% were from bachelor’s or higher degrees of education [25]; in the India study, 89.4% were from bachelor’s or higher degrees of education [27]; in the Nigeria study, all participants were from high school or a higher level of education [26]; and in the Malaysia study, 99.4% were from secondary or tertiary education [24].

Our multivariate analysis indicated that individuals with more accurate knowledge and more positive attitudes had more frequent best practices than their counterparts. These findings corroborate previous studies in South Korea, China, Malaysia, Nigeria, and Peru, which reported that sufficient knowledge is necessary for more positive attitudes and more frequent best practices during the pandemic [24,25,26,28,29]. However, the findings contradict a study in Ecuador, Iran, and Bangladesh, and a prior study in Indonesia, both of which found discrepancies in this area [4,23,30,31,32]. For example, greater knowledge of COVID-19 is insufficient to change individual attitudes toward COVID-19 prevention, according to the Ecuadorian KAP survey [4], which found that most of respondents from graduate and post-graduate education believe that greater knowledge of COVID-19 is insufficient to change individual attitudes toward COVID-19 prevention. Another KAP study involving Indonesian graduate students found that knowledge of COVID-19 was not consistent with positive attitudes or frequent COVID-19 preventative actions [23]. This discrepancy is interesting given the specific contexts and study population of our study. Hence, the discrepancies of the findings suggest that attitudes, which are related to individual beliefs rather than education, are a primary motivator for action regarding threats to health [33,34,35]. That is, when people believe that success is likely, they are more likely to act. For example, individuals need to believe that washing their hands and wearing masks will protect them from infection, beyond merely being informed that they should engage in and maintain these behaviors. In order for residents to engage in prudent behavior after receiving information, they need to believe that the practices suggested will be effective.

Our study also revealed that females had more accurate knowledge, more positive attitudes, and more frequent best practices than males. These findings confirm previous studies in Saudi Arabia, Palestine, Malaysia, China, and Indonesia, which reported that women had more accurate knowledge and were more engaged in collecting health information regarding COVID-19 prevention [7,23,24,36,37,38,39]. For example, a KAP survey in Malaysia indicated that 57.9% of women had more accurate knowledge than 42.1% of men [24], whereas 62.4% of women had more accurate knowledge than 37.6% of men in this study. This finding also corroborates well-documented evidence that women assume an important role in looking after the health of their families and, therefore, that they more often seek accurate healthcare information than their male counterparts [40,41].

Earlier studies in Malaysia, Saudi Arabia, Palestine, and Indonesia have also shown that senior adults are more likely to have more accurate KAP regarding COVID-19 prevention than young adults [24,38,39,42]. As senior adults have more health concerns and needs, they are more likely to seek and find accurate health information than young adults [43]. However, our findings in the present study were inconsistent. Middle-aged and senior adults had more accurate knowledge than young adults, while a non-significant association was detected for older adults on attitudes and practices. These discrepancies could be attributed to a lack of capability of older adults to implement good preventive practices in rural and resource-limited settings such as in Malang. Older population characteristics in previous KAP research in Malaysia, Saudi Arabia, Palestine, and Bangladesh were largely from a high school education or higher level [24,38,39,42,44], whereas older population characteristics in this current study were mostly from an elementary education or lower level. In addition, most older adults in Malang often do not have access to health facilities and sources [45], while prior KAP research in Malaysia, Saudi Arabia, Palestine, and Bangladesh focused on urban settings, where senior persons often have easier access to health facilities and resources.

Formal education represents individuals’ access to learning and obtaining knowledge [45]. Thus, we expected individuals with more formal education to have more accurate knowledge, a more positive attitude, and more frequent best practices than those with less formal education. Our study confirms prior KAP studies in Bangladesh, Malaysia, South Korea, China, Saudi Arabia, India, and France which documented a positive association between formal education and KAP [3,7,24,27,37,38,39,46,47]. A KAP study in Bangladesh reported among respondents who were identified as having more accurate knowledge that 87.6% of them held bachelor’s or master’s degrees [3], while in this current study we found 77.2% of those who had more accurate knowledge were individuals with graduate or post-graduate degrees.

Marital status captures the social and economic supports in which married individuals often have more support than widowed and divorced individuals [48]. Studies also identified that divorced individuals and widowed women are often linked to reduced access to health resources and information [49]. Accordingly, the gaps in knowledge among divorced and widowed individuals were also demonstrated in this study, supporting prior studies in Malaysia, Nigeria, Saudi Arabia, and Palestine [24,26,39,50]. In addition, gaps in attitudes and practices were shown between widowed and married individuals. These findings may explain health behavior disparities between those groups, as in prior studies in Malaysia, India, Bangladesh, Pakistan, Singapore, and Europe [3,24,27,30,51,52,53,54]. However, single individuals were less likely to engage in frequent best practices than married individuals, which is not surprising given that most single individuals are young adults with less accurate knowledge and less positive attitudes regarding the pandemic [24].

Family income portrays a household’s economic status, which often relates to family capabilities to access health resources and facilities, including health information [55]. Therefore, individuals from higher-income families are expected to have more accurate knowledge, more positive attitudes, and more frequent best practices than individuals from lower-income families. However, this study contrasted with studies in China, Bangladesh, and the Philippines, which revealed that family income relates to more accurate knowledge and better practices and attitudes toward COVID-19 prevention [3,25,56]. For example, a KAP study in Bangladesh reported among those who have more accurate knowledge that 73.5% came from a middle- or higher-income family [3], while in our study we found that 86.9% of respondents who reported having more accurate knowledge were from lower-income families. A possible explanation for this discrepancy might be the perceived risk among wealthier families [57]. With their greater economic resources, wealthier families may perceive lower risks of infection than poorer families. Another possible explanation for these discrepancies may relate to the settings and populations of the current study in which most richer families come from a low level of educational background. Prior KAP research in China, Bangladesh, and the Philippines, on the other hand, found a linear relationship between economic position and educational level [3,25,56].

Type of employment often relates to social and economic status, which may also be associated with knowledge, attitudes, and practices regarding COVID-19 prevention [58]. This study showed inconsistent associations between the type of employment and knowledge, attitudes, and practices. As expected, farmers and traders demonstrated less accurate knowledge than students. The results also revealed that traders engage in less positive attitudes and less frequent best practices than students. Moreover, the null findings of most employment types may explain the lack of gaps between students and other individuals with different employment backgrounds in terms of knowledge, attitudes, and practices. This finding confirms earlier studies in Bangladesh, the Philippines, China, and India, which reported similar results [3,25,27,56]. As most of the students in this study are young adults, the null findings may also reflect the lack of young adults’ concerns and their adherence to controlling the pandemic in Malang. A KAP study in Bangladesh also reported a similar issue and highlighted that young adults’ attitudes concerning COVID-19 are critical for their adherence to the government’s control actions to break the chain of contamination [3]. Recent studies focusing on vaccine hesitancy also found that vaccine hesitancy among younger ages is higher than in senior ages and this hesitancy led to lower uptake of COVID-19 vaccination among young people [59]. Therefore, how the government and communities deliver effective health promotion strategies to the groups to influence their behaviors is critical to increase vaccine coverage.

The associations between family type and knowledge, attitudes, and practices regarding COVID-19 prevention was consistent in the present study. Individuals who lived in joint families had more accurate knowledge, more positive attitudes, and more frequent best practices than those who lived in nuclear families. These findings also arose in studies in Bangladesh, Saudi Arabia, India, and Pakistan [3,6,27,30,60].

Residence in an urban or rural area often determines individuals’ access to health care, including health information. Our study revealed a null association of place of residence with knowledge about COVID-19. These findings are consistent with studies in South Korea, Ecuador, Saudi Arabia, Malaysia, and Singapore [4,5,28,38,51], but they contrast with results in China, Bangladesh, India, and Pakistan [4,7,27,60]. These discrepancies may relate to a particular region’s characteristics. We found that individuals living in rural areas engaged in less positive attitudes and less frequent best practices than those living in urban areas. These results are likely related to the characteristic of population density in rural areas and farmers’ knowledge of COVID-19 prevention, while prior studies mostly focused on urban areas with a large percentage of samples from a higher education level [4,7,27,60].

During the pandemic, community health workers have been playing a key role in health promotion programs [61]. The Indonesian government trains community health workers to disseminate useful information regarding COVID-19 prevention to local residents. In this study, these benefits were shown by the positive relationship between knowledge, attitude, and practices and being informed about COVID-19 by these workers. Our findings support previous studies in India, Botswana, Brazil, and Indonesia that highlight community health workers as the backbone of these countries’ healthcare systems [62,63,64,65]. While these studies reported on the benefits of community health workers for various health services during typical times, our study demonstrates that community health workers also play a pivotal role in fighting the pandemic in Indonesia [62,63,64,65].

As with many other governments, the pandemic has led the Indonesian government to use mobile technology for tracing COVID-19 and disseminating useful information regarding COVID-19 prevention. Among the mobile technology that Indonesians have used widely is JKN mobile health technology sponsored by Indonesian Social Security Administrator for Health (BPJS), which was designed to support the implementation of universal health insurance program. The present study found that individuals who used the app had more accurate knowledge, more positive attitudes, and more frequent best practices. These findings may indicate the benefits of the mobile health app for improving knowledge, attitude and practices among its users. Such benefits have also been reported in prior studies in Saudi Arabia, China, Singapore, and India [10,51,66,67,68]. For example, a study of digital contact tracing in Singapore found that the digital app is essential to control the pandemic in the absence of effective treatment and vaccines.

Some limitations of this study must be acknowledged. First, the cross-sectional design used in this study made it impossible to address causality. Hence, we must see the findings as associations rather than as indicating causality. Second, our study did not explore the associations of perceived barriers such as beliefs, culture, or information-seeking behaviors, with individual knowledge, attitudes, and practices in COVID-19 prevention [31]. Future studies should explore how those sociocultural and communication factors explain COVID-19 knowledge, attitudes, and practices among the general public.

Despite these limitations, our study provides valuable insights for global health proposal in order to improve health behaviors among the general public in the context of the scarcity of health resource settings. First, our study highlights that knowledge can play a crucial role in enhancing the public practice of preventive behaviors during a pandemic. Therefore, effective dissemination of health information about COVID-19 is needed to slow and control the pandemic. Our findings also indicate that mobile health technology and the presence of community health workers are associated with individuals’ knowledge and preventive behaviors regarding COVID-19. Given the fact that there are more than 3.5 billion smartphone users worldwide [68], policymakers can develop strategies to increase the use of mobile health technology in order to address COVID-19 knowledge disparities in communities. For example, through the mobile health app, the government and community can not only disseminate various information regarding the pandemic, but also can monitor and track infection chains, provide rapid support and information in the event of an illness or contact with an infected individual, and assist people in quarantine by monitoring their health and adapting information to preventive action.

Furthermore, our findings also suggest that community health workers, particularly in low-income countries with weak health systems, are positioned to play a critical role in combating the epidemic. The government can deploy community health workers to carry out effective public health education strategies. Community health workers who are properly equipped, trained, and supported as part of a well-functioning health system can aid in containing the epidemic. For example, because community members trust community health workers, allowing them to disseminate information and combat COVID-19 disinformation in the community may be more effective. Notably, this study discovered a high prevalence of misunderstanding about the source of infection through eating or contact with wild animals, with just 30.3% of respondents accurately responding that the information was incorrect.

Second, our study shows disparities in knowledge about COVID-19 in which those with less accurate knowledge tend to be males, young adults, divorced and widowed individuals, and people who are less educated, live in a nuclear family, are not informed by community health workers, and do not use mobile health technology. We documented that these disparities were also revealed within prior KAP surveys especially in low-middle income countries. Hence, our findings suggest that public health education strategies may prioritise these groups as they are more vulnerable in the pandemic in those countries.

## 5. Conclusions

KAP concerning COVID-19 in Malang District was low, which reflects the KAP level of the general population in Indonesia. Compared to the KAP level of the general population in other developing countries such as Malaysia, Bangladesh, India, and Nigeria, this study uncovered markedly reduced accurate knowledge, positive attitudes, and frequent best practices concerning KAP in Indonesia. The findings suggest some global health proposals to implement effective health promotion and prevention strategies in order to improve health behavior among the general public, especially in a country with a significant scarcity of health facilities.

## Figures and Tables

**Figure 1 ijerph-19-04287-f001:**
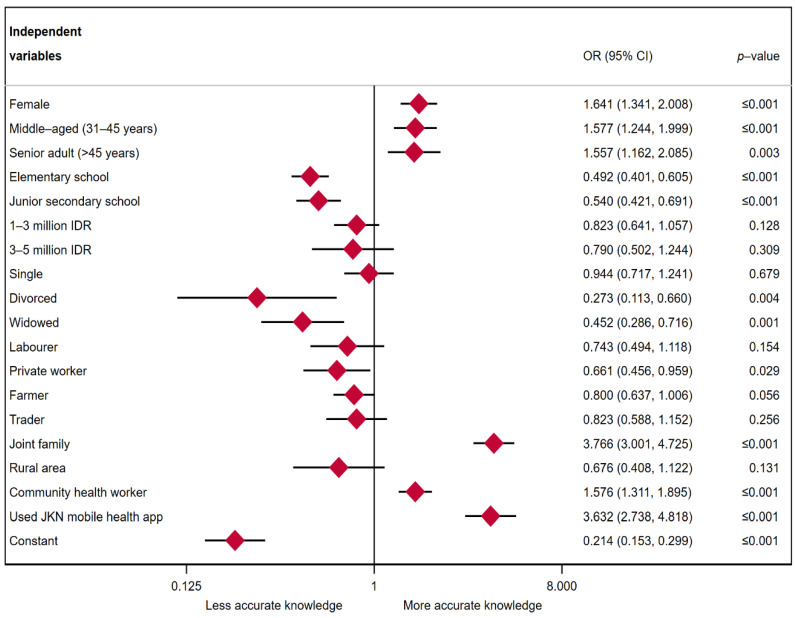
Multiple logistic regression results: knowledge about COVID-19.

**Figure 2 ijerph-19-04287-f002:**
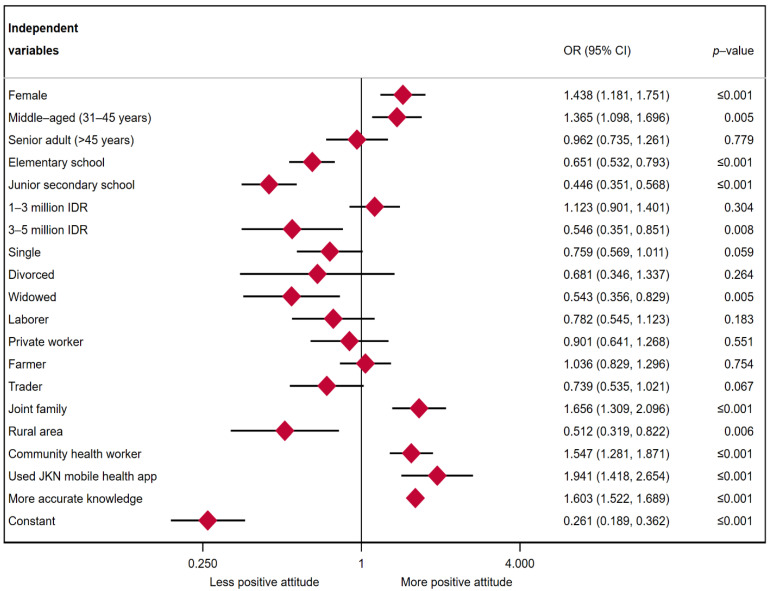
Multiple logistic regression results: attitudes towards COVID-19 prevention.

**Figure 3 ijerph-19-04287-f003:**
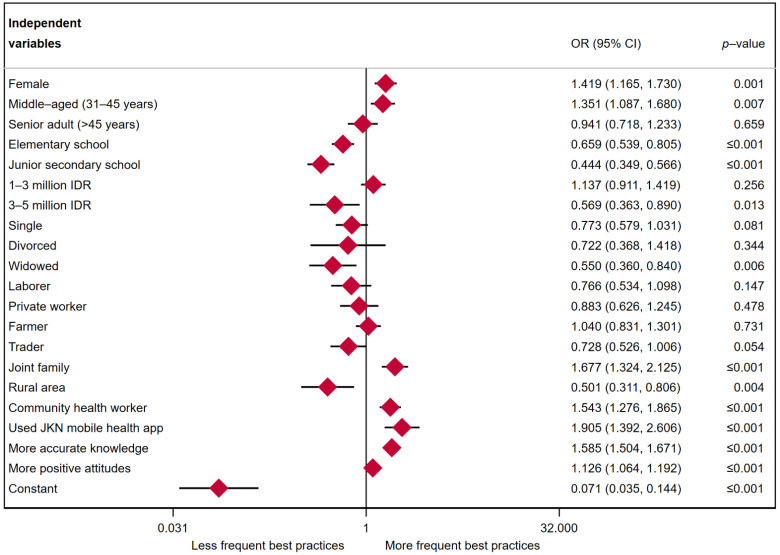
Multiple logistic regression results: practices toward COVID-19 prevention.

**Table 1 ijerph-19-04287-t001:** Questionnaires on KAP regarding COVID-19 prevention.

No	Knowledge about COVID-19	Coding and Cut-Off Levels
1	Is COVID-19 a dangerous disease?	Correct answer = 1, wrong answer or don’t know = 0. A cut-off level of ≥4 was chosen to indicate a respondent with a precise understanding of COVID-19 [4].
2	Does it affect only humans?	
3	Does it transmit from humans to animals?	
4	Does it transmit from animals to humans?	
5	Is it transmitted by animal products (e.g., milk, meat)?	
6	Is it transmitted in well-cooked products?	
	**Attitudes toward COVID-19**	
1	It is crucial to report a suspected case to health authorities	Disagree = 0, undecided = 1, agree = 2. A cut-off level of ≥11 was set to indicate a respondent with positive attitudes towards the prevention of COVID-19 [4].
2	It is important to use a face mask in crowded places	
3	It is important to wash hands and face after being outside	
4	COVID-19 is a preventable disease	
5	It can be treated at home	
6	Health education can play an important role in COVID-19 prevention	
	**Practice toward COVID-19**	
1	Do you use tissues or handkerchiefs when you cough/sneeze?	Yes = 1, no = 0, sometimes = 0 for questions 1–6. For question 7, yes = 0, sometimes = 0, no = 1. A cut-off level of ≥6 was set to indicate frequent best practices [4].
2	Do you wash hands frequently using water and soap?	
3	Do you avoid touching face and eyes?	
4	Do you maintain social distance (or quarantine at home)?	
5	Do you eat healthy food focusing on the outbreak?	
6	Do you maintain a healthy lifestyle focusing on the outbreak?	
7	Do you obey all government rules related to COVID?	

**Table 2 ijerph-19-04287-t002:** Sociodemographic characteristics and KAP of study participants (*N* = 3425).

Variables	*n* (%)
**Sex**	
Male	1286 (37.6)
Female	2139 (62.4)
**Age**	
Young adult (17–30 years)	1146 (33.5)
Middle-aged adult (30–45 years)	1565 (45.7)
Older adult (>45 years)	714 (20.8)
**Education**	
Elementary or less	1763 (51.5)
Junior secondary	699 (20.4)
High school or more	963 (28.1)
**Monthly family income (in IDR)**	
<1 million	2818 (82.3)
1–3 million	485 (14.1)
>3 million	122 (3.6)
**Marital status**	
Single	736 (21.5)
Married	2508 (73.2)
Divorced	41 (1.2)
Widowed	140 (4.1)
**Type of employment**	
Student	948 (27.7)
Laborer (i.e., construction, factory workers)	205 (6.0)
Civil servant	62 (1.8)
Privately employed (i.e., bank, supermarket, restaurant, hotel)	230 (6.7)
Farmer	1763 (51.5)
Trader (merchant, dealer, salesperson)	217 (6.3)
**Family type**	
Nuclear	2185 (63.8)
Joint	1240 (36.2)
**Used JKN COVID-19 screening app**	
No	3179 (92.8)
Yes	246 (7.2)
**Informed by community health workers about COVID-19 prevention**	
No	2596 (75.8)
Yes	829 (24.2)
**Residential area**	
Rural	3325 (97.1)
Urban	100 (2.9)
**Knowledge about COVID-19**	
More accurate	914 (25.3%)
Less accurate	2661 (74.7%)
**Attitudes towards COVID-19**	
More positive attitude	784 (36.6%)
Less positive attitude	2641 (62.4%)
**Practices regarding COVID-19**	
More frequent best practices	1670 (48.8%)
Less frequent best practices	1755 (51.2%)

**Table 3 ijerph-19-04287-t003:** Knowledge, attitudes and practices regarding COVID-19.

Knowledge about COVID-19	Yes	No	Don’t Know
Is COVID-19 a dangerous disease?	3251 (94.9%)	84 (2.5%)	90 (2.6%)
Does it affect only humans?	2484 (72.5%)	586 (17.1%)	355 (10.4%)
Does it transmit from humans to animals?	1133 (33.1%)	1063 (31.0%)	1229 (35.9%)
Does it transmit from animals to humans?	1608 (46.9%)	640 (18.7%)	1177 (34.4%)
Is it transmitted by animal products (e.g., milk, meat)?	1039 (30.3%)	1116 (32.6%)	1270 (37.1%)
Is it transmitted in well-cooked products?	730 (21.3%)	1696 (49.5%)	999 (29.2%)
**Attitudes toward COVID-19**	**Agree**	**Undecided**	**Disagree**
It is crucial to report a suspected case to health authorities	3209 (93.7%)	159 (4.6%)	57 (1.7%)
It is important to use a face mask in crowded places	3328 (97.2%)	62 (1.8%)	35 (1.0%)
It is important to wash hands and face after being outside	3305 (96.5%)	63 (1.8%)	57 (1.7%)
COVID-19 is a preventable disease	3151 (92.0%)	211 (6.2%)	63 (1.8%)
It can be treated at home	2388 (69.7%)	532 (15.5%)	505 (14.7%)
Health education can play an important role in COVID-19 prevention	3198 (93.4%)	140 (4.1%)	87 (2.5%)
**Practices regarding COVID-19**	**Yes**	**No**	**Sometimes**
Do you use tissues or handkerchiefs when you cough/sneeze?	1983 (57.9%)	231 (6.7%)	1211 (35.4%)
Do you wash your hands frequently using water and soap?	2470 (72.1%)	66 (1.9%)	889 (26.0%)
Do you avoid touching face and eyes?	1774 (51.8%)	340 (9.9%)	1311 (38.3%)
Do you maintain social distance (or quarantine at home)?	1961 (57.3%)	371 (10.8%)	1093 (31.9%)
Do you eat healthy food focusing on the outbreak?	2123 (62.0%)	207 (6.0%)	1095 (32.0%)
Do you maintain a healthy lifestyle focusing on the outbreak?	2374 (69.3%)	112 (3.3%)	939 (27.4%)
Do you obey all government rules related to COVID-19?	1984 (57.9%)	590 (17.2%)	851 (24.8%)

## Data Availability

The datasets used and/or analyzed during the current study are available from the corresponding author upon reasonable request.

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
