# Peer review of "A Cross-Sectional Study of Knowledge, Attitudes, and Practices concerning COVID-19 Outbreaks in the General Population in Malang District, Indonesia"

_ijerph, 2022, doi:10.3390/ijerph19074287_

Round 1

Reviewer 1 Report

Authors wrote an interesting paper. Idea research is good and the setting of research is rare. Below my minor suggestions

  1. Introduction: updata data on burden of SARS CoV2 wordwilde and in Indonesia at the days of resubmission
  2. Methods and results: well presented.
  3. Discussion: compare better your data with other data in literature. Discuss also the role of young in vaccine adherence and a special role to control the epidemic. Furthermore, add some global health proposal that came from your interesting study.

Author Response

Reviewer 1 (green highlight in the revised manuscript)

Authors wrote an interesting paper. Idea research is good and the setting of research is rare. Below my minor suggestions

1. Introduction: update data on burden of SARS CoV2 worldwide and in Indonesia at the days of resubmission

Response: the data on burden of SARS CoV2 worldwide and in Indonesia were updated per resubmission date, 27 March 2022

2. Methods and results: well presented.

Response: Thank you for the positive comments.

3. Discussion: compare better your data with other data in literature. Discuss also the role of young in vaccine adherence and a special role to control the epidemic. Furthermore, add some global health proposal that came from your interesting study.

Response: We revised discussion section by comparing our data with other data from prior KAP surveys. We discussed the role of young to control the pandemic and in vaccine adherence based on our data and prior studies. We added some global health proposal in the implications of our study and conclusion.

Reviewer 2 Report

Thank you for asking me to review this article. I have read with pleasure the manuscript which offers numerous food for thought for the possible deepening of this theme also in other healthcare contexts.
The subject of study is very important both as regards the epidemiological aspects related to COVID-19 and as regards the influence of particular determinants (such as Knowledge, Attitudes and Behaviors) on the health outcomes or on the health choices of the population.
In this regard, since it is well known in literature that the decision-making process inherent to health choices is also strongly influenced by what you read online, it would have been interesting to consider this aspect as well and the possible influence that online information or disinformation exerts. in relation to the Knowledge, Attitudes and Behaviors of the investigated cohort. However, the manuscript is certainly interesting and is well described in content and form. I therefore believe that it can be published in its current form.

Author Response

Reviewer 2 (blue highlight in the revised manuscript)

Thank you for asking me to review this article. I have read with pleasure the manuscript which offers numerous food for thought for the possible deepening of this theme also in other healthcare contexts. The subject of study is very important both as regards the epidemiological aspects related to COVID-19 and as regards the influence of particular determinants (such as Knowledge, Attitudes and Behaviors) on the health outcomes or on the health choices of the population. In this regard, since it is well known in literature that the decision-making process inherent to health choices is also strongly influenced by what you read online, it would have been interesting to consider this aspect as well and the possible influence that online information or disinformation exerts. in relation to the Knowledge, Attitudes and Behaviors of the investigated cohort. However, the manuscript is certainly interesting and is well described in content and form. I therefore believe that it can be published in its current form.

Response: Thank you for all positive responses. We added issue of disinformation and provide suggestions based on the findings.

Reviewer 3 Report

The manuscript is well-written and cogent and I offer only minor recommendations for improvement: 

  1. The authors report the response rate as 95.1%. I am not sure if this could be considered the response rate as it is not clear if the 3600 "potential respondents" represent the people who were asked to participate or just the people who consented. This needs clarification.
  2. It is not clear why the questionnaire is described as semi-structured. The questions presented in Table 1  seem to be 3-point Likert scale type items.
  3. In the discussion, the authors compared and contrasted their findings with the findings of other KAP studies conducted in other countries. It would be interesting to know if these studies were conducted in similar populations and settings. This might help to explain difference in outcomes across some studies.
  4. Please correct the typo in line 251 where "higher" appears instead of "lower"
  5. This is minor - Lines 285-286 reference 2 studies in the text but provides 4 different citations in parentheses. 

Author Response

Reviewer 3 (yellow highlight in the revised manuscript)

The manuscript is well-written and cogent and I offer only minor recommendations for improvement: 

1. The authors report the response rate as 95.1%. I am not sure if this could be considered the response rate as it is not clear if the 3600 "potential respondents" represent the people who were asked to participate or just the people who consented. This needs clarification.

Response: We change “potential respondents” into “study sample” in the revised manuscript (line 96-97).

2. It is not clear why the questionnaire is described as semi-structured. The questions presented in Table 1 seem to be 3-point Likert scale type items.

Response: We change a semi-structure questionnaire into questionnaire in the revised manuscript (line 99)

3. In the discussion, the authors compared and contrasted their findings with the findings of other KAP studies conducted in other countries. It would be interesting to know if these studies were conducted in similar populations and settings. This might help to explain difference in outcomes across some studies.

Response: In the discussion, we added populations and settings of other KAP studies in other countries which we used to compare and contrast our findings.

4. Please correct the typo in line 251 where "higher" appears instead of "lower"

Response: We revised “higher” to “lower” in the revised manuscript (line 254).

5. This is minor - Lines 285-286 reference 2 studies in the text but provides 4 different citations in parentheses. 

 Response: We added Iran and Bangladesh in revised manuscript (line 304-305)